# Determinants of Suicidality in the European General Population: A Systematic Review and Meta-Analysis

**DOI:** 10.3390/ijerph17114115

**Published:** 2020-06-09

**Authors:** María Teresa Carrasco-Barrios, Paloma Huertas, Paloma Martín, Carlos Martín, Mª Carmen Castillejos, Eleni Petkari, Berta Moreno-Küstner

**Affiliations:** 1Department of Personality, Assessment and Psychological Treatment, University of Malaga, 29010 Malaga, Spain; maria.teresa.carrasco.barrios@gmail.com (M.T.C.-B.); palomamg1175@gmail.com (P.M.); mccasang@gmail.com (M.C.C.); bertamk@uma.es (B.M.-K.); 2Primary Care Center of Marquesado, Área Nordeste de Granada, 18512 Granada, Spain; med000261@hotmail.com; 3Social and Behavioural Sciences, European University Cyprus 6th Diogenous st., Nicosia 2063, Cyprus; E.Petkari@euc.ac.cy; 4Biomedical Research Institute of Malaga (IBIMA), 29010 Málaga, Spain

**Keywords:** suicidality, death wishes, suicidal ideation, suicidal plans, suicidal attempts, general population, risk factors

## Abstract

Close to one million people commit suicide each year, with suicidal attempts being the main risk factor for suicide. The aim of this systematic review and meta-analysis is to achieve a greater understanding of suicidality in the general population of Europe by studying associated factors and their statistical significance with suicidality, as well as the effect of the temporal moment in which suicidality is observed in a relationship. A search strategy was carried out in electronic databases: Proquest’s Psychology Database, Scopus, PsycINFO, Medline and Embase. Odds ratios (ORs), publication bias, influential studies on heterogeneity and analysis moderators were calculated. Twenty-six studies were included after meeting the inclusion criteria. Factors statistically associated with suicidality are female gender, age over 65 years, unemployment, low social support, adulthood adversity, childhood adversity, family history of mental disorder, any affective disorder, major depression, anxiety/stress/somatoform disorders, tobacco and substance use, any mental disorder and body mass index. As a limitation, a high heterogeneity between studies was found. Factors associated with suicidality in the general population are relevant for understanding the suicidal phenomenon.

Systematic review registration: PROSPERO (CRD42017075190).

## 1. Introduction

Every year, close to one million people commit suicide, affecting the contexts they belonged to and survivors [1]. According to the same source, there are ten to twenty more suicide attempts than suicides committed. In addition, suicide attempts are the main risk factor for suicide. For this reason, the study of suicidality is especially relevant, as their prevention decreases the risk of a completed suicide.

Suicidality has been conceptualized as a continuum that can progress from death wishes and tiredness of life to suicidal ideation, then to planning and attempts [2]. To understand the differences between the behaviors included in suicidality, the distinction established by Nock and Favazza [3] is useful. These authors define suicidal ideation as “thoughts of engaging in behaviors intended to end one’s own life”. In this way, they differ from death wishes, which are characterized by being passive. They consider suicide plans as “the [cognitive] formulation of a specific method through which one intends to die”. Finally, they define suicide attempts as “engagement in potentially self-injurious behavior in which there is at least some intent to die”, emphasizing the intention to die.

A greater understanding of suicidality involves investigating their associated factors [4]. Different studies have presented original data on the prevalence and factors associated with suicidality in recent decades, with an increasing number of systematic reviews, specifically meta-analyses, in recent years. These studies include systematic reviews and/or meta-analyses that have focused on investigating suicidality in the general population [5]; in certain specific populations such as, for example, children and adolescents [6], students [7], prisoners [8] and inpatients [9]; or combinations of different types of populations [10,11,12,13,14]. Factors that have been studied in systematic reviews and/or meta-analyses include individual psychological variables such as hopelessness [10], alexithymia [11] or self-esteem [12], and groups of psychological variables—for example, a systematic review that includes mental pain, communication difficulties, decision-making impulsivity and aggression [14]; mental disorders such as depression [10], affective disorders [15], anxiety disorders [16], obsessive compulsive disorder [17], substance use [18], psychosis [19], comorbid obsessive compulsive disorder and bipolar disorder [20], and any mental disorder [8,13]; medical problems and conditions such as inflammatory cytokines [21], concussion [22], psoriasis [23] and body mass index [24]; demographic factors such as gender [25], sexual minority [26], poverty [27], marital status [28], employment situation [29] and age [12,13,30,31,32], or various demographic factors together [14]; psychosocial factors such as adverse life experiences [33,34] and parental death or suicide [35]; parental psychopathology [36] and environmental factors [37,38], among other factors.

Systematic reviews and/or meta-analyses have been conducted in different specific geographic areas such as countries [5,39,40,41,42] or have included several countries. Some have also considered several factors, such as the systematic review of Nock et al. [43], which provided information about certain associated risk factors worldwide (psychological, psychiatric, biological and stressful life events); the international meta-analysis of Franklin et al. [44] that addresses suicidality, consummate suicide and associated risk factors (e.g., biology, cognitive variables and processes, demographics, psychopathology), but includes only longitudinal studies; the systematic review, but not meta-analysis, conducted by Cano-Montalbán and Quevedo-Blasco [45], which examines sociodemographic factors and their relationship with suicidality in the general population in Europe and America, and a meta-analysis that focuses on demographic factors as predictors of suicidality worldwide [46].

Thus, although a variety of demographic, psychosocial and clinical risk factors for suicidality have been investigated, the explanatory potential of these factors does not seem to be much greater than chance [44]. This observation was recently checked to determine whether it occurs with the strongest predictors (e.g., previous suicidality, mental disorders, hopelessness), finding a similar result [10,19,47,48]. According to the authors, these results may be related to the lack of systematization regarding the type of population included, as some studies are limited to one type of population, while others combine various types: differences in methodological designs, geographical areas, cultural factors considered and period of time in which suicidality is studied, as well as other factors, such as considering risk factors in isolation or in combination [49]. As Lagares-Franco et al. [50] comment, the lack of systematization when measuring the frequency of the presentation of suicidality may be another reason that explains the difficulty of studying suicidality, alongside the result that the methodological and terminological variability makes it difficult to make comparisons between studies.

Taking into account the complexity of suicidality as well as the differences that cultural and racial factors generate in such behaviors [51,52], we believe that studies which focus on similar societies offer more reliable results for each population studied. As an example, the study of Mathy [53] finds a relation between suicide attempts and sexual orientation in males for all continents except Europe. This difference between continents could also occur among factors associated with suicidality. Moreover, it is especially relevant to study the relationships between factors associated with suicidality depending on the time at which suicidality is observed.

To the best of our knowledge, in recent years, there have been no systematic reviews or meta-analyses on associated factors specifically in the European general population. Specific factors have been considered in European countries [42], but no attempt has been made to consider a wide range of factors in various European countries together, nor have studies differentiated between periods of time of suicidality. This type of study is relevant because to prevent suicide, a serious problem today [1], it is necessary to learn more about the continuum of suicidality. Moreover, the study of these factors in the general population enables larger-scale prevention. In addition, as suicidality generates discomfort, prevention would translate into greater wellbeing in the general population.

Given this background, the objective of this systematic review and meta-analysis is to achieve a greater understanding and a closer approach to the knowledge of suicidality in Europe by studying associated factors and their statistical significance with these behaviors. Likewise, we differentiate between the period time in which suicidality occurred. Our intention is to include the studies under in the World Mental Health (WMH) Survey Initiative as well as those that follow the same methodology for studying suicidality in the general population, within the WHO Europe Region. For this purpose, we took the Nock et al. [43] study as a reference, which analyses studies on prevalence and associated factors worldwide, including Europe. Our purpose is to continue this review, and update the information available in the European Region published within the following ten years, similar to other authors that have studied suicidality in other regions [5,39,54]. As a result, the question we intend to solve in this study is: are the factors considered and suicidality significantly associated in the general population in Europe? This systematic review and meta-analysis is part of a larger study, which first aims to investigate the prevalence of suicidality in Europe [55]. In line with this topic, our research group has recently published a study focusing on the prevalence and factors associated with suicidality in the general population in Andalusia (southern Spain) [56].

## 2. Materials and Methods

### 2.1. Search Strategy

The search strategy was the same for both systematic review studies previously noted, which was carried out following the PRISMA statement [57] (Appendix A).

An initial broad systematic review was conducted. We initiated a search of published research using the following electronic databases related to social and medical sciences: Proquest’s Psychology Database, Scopus, PsycINFO, Medline and Embase. We used the following search string: Title (suicid*) AND Abstract ((prevalence OR epidemiolog*) OR (“risk factor*” OR “associated factor*” OR “correlated factor*”)). The potential records involved literature published between January 1, 2008 and December 31, 2017.

This meta-analysis has been registered in PROSPERO (CRD42017075190).

### 2.2. Inclusion Criteria

The inclusion criteria were studies that (i) performed an analysis of data from the general population; (ii) were original studies providing primary data; (iii) were conducted in Europe; (iv) reported the prevalence of any suicidality (wishing to be dead, suicidal ideation, suicide planning and attempted suicide); (v) assessed at least one risk factor for any of these outcomes; (vi) covered a wide age range, i.e., adolescents to elderly people; (vii) were published between 2008 and 2017; (viii) had the full text available in English or Spanish.

As exclusion criteria, we considered articles that studied completed suicide and studies that did not assess any associated factor, as well as articles that did not offer sufficient results for comparison. Furthermore, we excluded studies of specific populations, such as inpatients, university students and members of the armed forces.

### 2.3. Selection Process

Two independent reviewers carried out the selection process, resolving the discrepancies through a third reviewer. The number of potentially eligible records identified through electronic searching was 18,287. After removing 11,358 duplicated articles, two independent reviewers screened the remaining 6929 records by applying the inclusion criteria to the titles and abstracts. As a result, 75 papers were reviewed by reading the full manuscript, of which 49 were eliminated due to including completed suicide but not suicidality (*n* = 2); data not being available (*n* = 12); being the only study that measured a certain factor, and thus being eliminated as it could not be compared with other studies (*n* = 1); not dealing with the general population (*n* = 5); not being conducted in Europe (*n* = 25); displaying duplicated data (*n* = 2) and the full text not being available (*n* = 2). Finally, a total of 26 studies on associated factors for suicidality were included in the present study (Figure 1).

### 2.4. Data Extraction

Data for the meta-analysis were extracted directly from the articles or, if they were not displayed directly, were calculated using the data provided. Two independent investigators carried out the data extraction, resolving the discrepancies through a third investigator. The variables collected from each of the different studies included in this review are detailed below. For more information on the variables included see Appendix A.

Citation-level variables: author/s and year;Methodological variables: country, age range of the sample and quality rating assessed with an adapted version of the Quality Assessment Tool for Quantitative Studies [58] (Appendix A) and instruments used to assess factors: assessment tools for demographic factors, assessment tools for psychosocial factors and assessment tools for clinical factors;Outcome variables: all types of suicidality and each type of suicidality (death wishes, suicidal ideation/thought/thinking, suicidal plans, suicidal attempts). For coding we used the criterion of including in each category the most severe suicidality type that was collected. For example, studies which combine ideation and attempted suicide were coded as suicidal attempts. As part of the dependent variable, the period of time when the suicidality occurred was collected: a specific point, 12-month period and lifetime, with point referring to the previous week, previous two weeks or previous month;Factors analyzed in the meta-analysis: demographic factors (gender, age up to 35 years, age between 35 and 65 years, age over 65 years, relationship status, residential setting, nationality, education, employment situation), psychosocial factors (social support, adulthood adversity, childhood adversity) and clinical factors (family history of mental disorder, any affective disorder, major depression, anxiety/stress/somatoform disorders, substance use, frequent alcohol consumption, tobacco use, any mental disorder, body mass index);Data for the meta-analysis were calculated with 2 × 2 tables showing the number of participants with and without the factor and the number of participants with suicidality with and without the factor. If any of these data were not provided, they were calculated by the authors using the data given in the article.

### 2.5. Quality Assessment

An adapted version of the Quality Assessment Tool for Quantitative Studies was used to assess the quality of the studies included in the review [58]. This evaluation tool evaluates eight different factors, which are the study design, analysis, withdrawals and dropouts, data collection practices, selection bias, intervention integrity, blinding as part of a controlled trial and confounders. However, the overall assessment for the original tool does not consider the categories of analysis and intervention integrity, so, in this study, we did not consider them either. Likewise, the confounders category and the blinding category, aimed at experimental studies, do not apply to the studies in this review, while the withdrawals and dropouts category was only taken into account for the lone study that included follow-up. The quality of each of the factors was evaluated using a three-level scale: “strong”, “moderate” and “weak”. In the present meta-analysis, the assessment was performed by three reviewers. See Appendix A for more information on the adjustments made.

### 2.6. Data Analysis

All analyses were performed using the statistical package R commander. For each of the potential associated factors, a contingency table was constructed and, when necessary, the odds ratio (OR) was obtained for each study included in the analysis. Subsequently, the pooled ORs were calculated for all studies with their corresponding 95% confidence intervals (CI). The method used to estimate tau square was the restricted maximum likelihood (REML). We used the random effects model, under the assumption that the studies included in the meta-analysis were conducted in a variety of populations that may differ from each other. Statistical significance was considered with a *p*-value < 0.05.

Statistical heterogeneity was evaluated using Cochran’s Q statistic, its *p*-value and Higgins and Thompson I^2^, in which a value less than 30% indicates that there is low heterogeneity, 30% to 50%, moderate and more than 50% to 100%, severe [59]. To analyze heterogeneity between studies, sensitivity and meta-regression analyses were performed. The sensitivity analysis excluded studies with a high influence on heterogeneity. A meta-regression was performed for several factors to check the influence of the period of time and the type of suicidality on heterogeneity. The R^2^ statistic that determines the percentage of variability explained by the model was calculated.

To detect publication bias, a funnel plot was examined by visual inspection and by applying the Egger’s test. The Duval and Tweedie trim and fill procedure was also used, which is a funnel symmetry test. This procedure produces an adjusted clustered effect size after taking into account missing studies due to publication bias.

## 3. Results

### 3.1. Characteristics of Studies

After the selection process explained above, a total of 26 studies were selected for inclusion in this systematic review and meta-analysis, as shown in Figure 1. The number of studies that included the different European countries from which data are analyzed are: three from Belgium, one from Bulgaria, four from England, one from Finland, two from France, five from Germany, three from Great Britain, two from Greece, two from Northern Ireland, two from Italy, one from Latvia, two from The Netherlands, one from Portugal, one from Romania, four from Spain, one from Sweden and two from Turkey. All the studies were cross-sectional, with the exception of one, which was a cohort. Regarding the suicidality type, two studies assessed death wishes, 22 studies assessed suicidal ideation, six studies assessed suicidal plans and 20 studies assessed suicidal attempts. Concerning the period of time of suicidality, nine studies assessed a specific point, ten assessed a 12-month period and 18 assessed lifetime (Table 1).

As associated factors, 21 studies assessed demographic factors, seven assessed psychosocial factors and 12 assessed clinical factors. In detail, concerning demographic factors, 19 studies assessed gender, five assessed age up to 35 years, two assessed age between 35 and 65 years, three assessed age over 65 years, ten assessed relationship status, seven assessed residential setting, two assessed nationality, three assessed education and six assessed employment situation. Concerning psychosocial factors, two studies assessed social support, three assessed adulthood adversity and four assessed childhood adversity. Finally, concerning clinical factors, four studies assessed family history of mental disorder, four assessed any affective disorder, four assessed major depression, six assessed anxiety/stress/somatoform disorders, five assessed substance disorders, three assessed frequent alcohol consumption, three assessed tobacco use, 11 assessed any mental disorder and two assessed body mass index (Table 1).

With regard to the quality assessed using an adapted version of the Quality Assessment Tool for Quantitative Studies, most studies had a moderate quality. One had a weak quality and three were rated as strong, so the remaining 22 studies showed a moderate quality (Table 1, Appendix A).

### 3.2. Meta-Analysis of Outcomes

#### 3.2.1. All Types of Suicidality

Certain factors had significant OR within demographic, psychosocial and clinical factors when considering all types of suicidality together, as well as all time periods (Table 2). Thus, for the demographic factors there were significant OR for gender, with being a woman presenting a higher risk for suicidality; age over 65 years presenting a lower risk compared to other ages; relationship status, with a higher risk when not in a stable relationship; and the employment situation, with being inactive presenting a higher risk. All psychosocial factors had significant OR: low social support, adversity in adulthood and adversity in childhood. All these psychosocial factors presented a greater OR than the demographic factors. In clinical factors, almost all presented significant OR: family history of mental disorder, any affective disorder, major depression, anxiety/stress/somatoform disorders, substance use, tobacco use, any mental disorder and body mass index. Only frequent alcohol consumption did not present significant OR. In general, the OR values of clinical factors were higher than those of the demographic factors. Major depression, any affective disorder and anxiety/stress/somatoform disorders stood out as factors with greater OR. Between-study heterogeneity in all cases was high, being greater than 75% in the majority of cases. The Forest plot for each factor is shown in Appendix A.

#### 3.2.2. Death Wishes

Considering death wishes in all time periods, the clinical factors of anxiety/stress/somatoform disorders (*n* = 4, OR = 2.38, 95% CI 1.08–5.24, *p* < 0.05) and any mental disorder (*n* = 6, OR = 3.03, 95% CI 1.60–5.75, *p* < 0.05) reached statistical significance. Relationship status (*n* = 2, OR = 1.03, 95% CI 0.43–2.42, *p* = 0.9524) was not statistically significant. Heterogeneity between studies was, in all cases, higher than 64%. For relationship status it was I^2^ = 92.80%, for anxiety/stress/somatoform disorders it was I^2^ = 64.43% and for any mental disorder it was I^2^ = 68.75%.

#### 3.2.3. Suicidal Ideation

As detailed in Table 3, the three categories of factors presented significant OR for certain factors when analyzing suicidal ideation in any time. For the demographic factors, a greater risk was found for female gender, under 35 years of age, under 65 years of age, when not in a stable relationship and occupationally inactive. For psychosocial factors, a greater risk was found for low social support and childhood adversity and for clinical factors, a family history of mental disorder, any affective disorder, major depression, anxiety/stress/somatoform disorders, substance use, tobacco use and any mental disorder. Demographic factors were those with lower OR, while major depression, any affective disorder and anxiety/stress/somatoform disorders were the factors with higher OR. Heterogeneity between studies was high, being in most cases higher than 74%.

#### 3.2.4. Suicidal Plans

The two demographic factors calculated had significant OR for any time period. These factors were being female (*n* = 16, OR = 1.78, 95% CI 1.41–2.25, *p* < 0.05) and over 65 years (*n* = 3, OR = 0.28, 95% CI 0.17–1.44, *p* < 0.05). Heterogeneity between studies was high for the first factor (I^2^ = 73.42%), while low for the second (I^2^ = 0%).

#### 3.2.5. Suicidal Attempts

The OR calculated for suicide attempts can be seen in Table 4. Regarding suicide attempts at any time, the following factors presented significant OR: female gender, age between 35 and 65 years and occupationally inactive as demographic factors, low social support, adversity in adulthood and in childhood as psychosocial factors and any affective disorder, major depression, anxiety/stress/somatoform disorders, substance use, tobacco use, any mental disorder and a high body mass index as clinical factors. As in others types of suicidality, clinical and psychosocial factors presented greater OR than demographic factors. Major depression and any affective disorder had the highest OR, followed by adversity in adulthood, adversity in childhood and anxiety/stress/somatoform disorders. In general, the heterogeneity between studies was high, although with greater variability than in the other suicidality.

### 3.3. Heterogeneity Analysis

A sensitivity analysis was used to verify the existence of influential studies on heterogeneity, as well as a meta-regression in 15 factors. Demographic factors analyzed with these methods were gender, age over 65 years, relationship status, residential setting, education and employment situation. Psychosocial factors were social support, adulthood adversity and childhood adversity. Clinical factors were any affective disorder, major depression, anxiety/stress/somatoform disorders, substance use, tobacco use and any mental disorder.

#### 3.3.1. Sensitivity Analysis

The most influential studies on heterogeneity, withdrawal of which causes a 5% reduction in heterogeneity, are shown in Appendix A. This analysis was performed taking into account all time periods and indicates the OR before and after removing the study, as well as the variation in heterogeneity.

In summary, the most influential studies were the study of Bebbington et al. [67], for gender in relation to all types of suicidality and suicidal ideations; the study of Boyd et al. [71], for gender in relation to suicidal plans and suicidal attempts for data from Portugal and from The Netherlands, respectively; Hiswåls et al. [80], for education in relation to all types of suicidality; Bruffaerts et al. [72], for adulthood adversity in relation to all types of suicidality; Gisle & Van Oyen [77] for social support in relation to all types of suicidality; Michal et al. [86], for any affective disorder in relation to all types of suicidality and for substance use in relation to suicidal ideation because of data referring to alcohol consumption; and Atay et al. [65], for any mental disorder in relation to death wishes due to data on major depression.

#### 3.3.2. Moderator Analysis

A meta-regression was carried out, including, as moderating variables, the time period in which the suicidality occurred and, in the case of all kinds, each type of suicidality.

In the case of the type of suicidality, the estimated effect for each factor remained consistent regardless of the specific type of suicidality. In relation to time period, it was observed that this moderator does explain part of the heterogeneity observed in certain combinations of factors and types of suicidality. The demographic and psychosocial factors in which this moderator was relevant were gender, age over 65 years, education, employment situation and social support in all cases when compared with a 12-month period, with the exception of gender; in relation to gender, education and social support when compared with a lifetime period (see Appendix A). The time period was relevant for all clinical factors analyzed: any affective disorder (point and lifetime), major depression (point and lifetime), anxiety/stress/somatoform disorders (point, 12-month period and lifetime), substance use (point and 12-month period), tobacco use (12-month period) and any mental disorder (12-month period). The 12-month period was not evaluated in the case of any affective disorder and major depression (see Appendix A).

When considering each type of NLCB, time period was analyzed as the moderator. In the case of suicidal ideation, the four factors evaluated presented significance for at least one level of the moderator (see Appendix A). These factors were gender (lifetime), anxiety/stress/somatoform disorders (point, 12-month period and lifetime), substance use (12-month period) and any mental disorder (12-month period). In suicidal attempts, substance use (12-month period) and any mental disorder (12-month period) were the only factors that reached significance (see Appendix A). This analysis could not be performed for death wishes due to a lack of data. In the case of suicide plans, only gender was analyzed for the 12-month period (OR = 0.64, 95% CI 0.25–1.64, *p* = 0.35) and for lifetime (OR = 1.84, 95% CI 1.44–2.35, *p* < 0.05), which was the reference in the analysis. The heterogeneity explained with R^2^ was 0%. 

A meta-regression after the sensitivity analysis was performed for gender in all types of suicidality and each category of suicidality, because it was the factor with the greatest number of effect sizes in our study and had a heterogeneity reduction greater than 9%. After these analyses, the significance seen in suicidal plans by time period disappeared. The rest of the data remained similar (see Appendix A).

### 3.4. Publication Bias

The analysis of publication bias using Egger’s test showed the existence of bias for eight combinations of factor and outcome: frequent alcohol consumption with suicidal ideation, and all suicidality with gender, age up to 35 years, anxiety/stress/somatoform disorders, frequent alcohol consumption, tobacco use and body mass index. A *p*-value < 0.05 was used as a reference to indicate publication bias. Using the trim and fill method, three continued to show publication bias: frequent alcohol consumption in all types of suicidality and suicidal ideation, and age up to 35 years in all types of suicidality. In addition, the education and age between 35 and 65 years factors in all types of suicidality showed publication bias. These results are shown in Appendix A.

## 4. Discussion

This study seeks to investigate what factors are associated with suicidality in the general population through a systematic review and meta-analysis of studies carried out in Europe. Within the main results, it was found that several factors studied are related to suicidality, that these relationships vary depending on the type of suicidality and that there are effects of the moderator period of time.

Among our results, we highlight the trend of the highest OR for clinical factors, followed by psychosocial factors and, finally, demographic factors. Depression and any affective disorder were the factors with the highest OR in all outcomes. Finally, the high OR obtained for the factors adversity in adulthood and adversity in childhood in suicide attempts are also remarkable.

When comparing our results with those obtained by other authors, the values calculated in several factors stand out. Among demographic factors, female gender showed similar OR to that found in other meta-analyses [46,102]. This can be explained by a more positive attitude towards the search for treatment for mental disorders and psychological problems in women than in men [103]. The results found for the age factor show that all subgroups showed differences with respect to the rest when considering different types of suicidality. Ages up to 35 years had a significant OR in suicidal ideation, ages between 35 and 65 years had a significant OR in suicidal attempts and ages over 65 years had a significant OR in all outcomes, but with ages over 65 years being a protective factor. This is an unexpected result since people over 65 have more functional disabilities, and it would be expected that functional disabilities had some role in the relationship between advanced age and suicidal behaviors [104]. Nevertheless, another meta-analysis [46] found that age did not present a significant OR in any of the suicidalities considered, in this case suicidal ideation and suicidal attempts. Relationship status obtained a significant OR in suicidal ideation and all suicidalities. In another meta-analysis that studied marital status, no association was found in either ideation or suicidal attempts, although in the case of ideation it was very close to significance [105]. Given these results, it is possible that the difference lies in the existence or not of a stable relationship, as assumed in our study, rather than in the marital state. In this way, having a stable partner would be associated with a lower risk of suicidal ideation. The residential setting factor did not obtain significant OR in any suicidality. With this factor, we differentiate between urban and rural residential area. In China, an OR was observed in the direction of showing more risk in those who live in a rural area than an urban area [5]. However, it is possible that this factor is not relevant in the European context. In addition, it may be mediated by other factors such as socio-economic level or vital adversities and difficulties. When considering all types of suicidality, suicidal ideation and suicidal attempts, employment situation proved to have a significant OR in the direction of risk for occupationally inactive. Another meta-analysis shows that neither ideation nor suicidal attempts show a significant association with the employment situation, but in the case of ideation the significance was very close [46]. Regarding education level and nationality, in no case were these factors significant, as in another meta-analysis that focuses on demographic factors worldwide [46]. It should be noted that both our study and the study cited encountered the problem of the shortage of effect sizes that analyzed these factors, which implies that making statements about them is complicated by the lack of studies. On the other hand, the aforementioned study conducted in China does find a significant relationship with the educational level [5]. This may indicate the relevance of sociocultural factors to the existence of an association.

All psychosocial factors were significant when considering all types of suicidality and suicidal attempts, but when considering suicidal ideation, only social support and childhood adversity were significant. Other studies show different results in some aspects. A meta-analysis conducted in adolescents and young adults found that childhood adversity was significant for suicidal attempts, with a similar OR to the obtained in our meta-analysis, whereas the OR was not significant for social support or adulthood adversity [106]. Two other meta-analyses focusing exclusively on childhood adversity coincide in finding a similar result to that found in our meta-analysis, although one that considered sexual abuse had an OR of 2.43 [107], while another found an OR of 3.78 in the general population when considering all types of adversity and suicidality [108]. Therefore, within this group of psychosocial factors, the factor that seems to remain clearly relevant for its association with suicidality is childhood adversity. The lack of meta-analyses analyzing social support stands out. The importance of this factor is pointed out in several systematic reviews in the direction of a possible association of a higher risk of all types of suicidality with low social support [109,110,111].

With regard to clinical factors, we obtained significant OR for family history of mental disorder when considering all types of suicidality and suicidal ideation. A meta-analysis found that a family history of mental disorder and alcohol or drug abuse was significant with respect to suicidal attempts [106], but there were no more results related to suicidality. Our study obtained an OR of 11.06 for major depression in suicidal ideation, while other studies found an OR close to two [10,44]. Regarding these results, it should be noted that suicidal ideation is one of the diagnostic criteria for major depression [112]. Significance was also achieved for suicide attempts, but with a lower OR than in suicidal ideation. The same pattern was obtained in both studies cited, although with a lower value of OR [10,44]. For any affective disorder, a significantly high OR was also found when considering all types of suicidality, suicidal ideation and suicidal attempts, as occurred in major depression. A meta-analysis carried out in young people found the same results for any affective disorder [13]. The same was found in a previous meta-analysis for bipolar disorder [15] and in a meta-analysis that focused on people with bipolar disorder and comorbid substance use disorders, including alcohol use disorder [113]. As for the factor anxiety/stress/somatoform disorders, the same pattern was found as in any affective disorder, adding a significant OR in death wishes. Thus, the OR, although lower in this case, was significant in all types of suicidality, death wishes, suicidal ideation and in suicidal attempts. In addition, the OR was calculated for death wishes, though it was not significant. While this result does not coincide with that obtained for suicidal attempts in a meta-analysis carried out in young people [13], another meta-analysis also found an association for both suicidal ideation and suicidal attempts [16]. The OR found was somewhat less than that in the present study. Use of substances and use of tobacco presented significant OR in all categories of suicidality analyzed. For frequent alcohol consumption, though, it was not significant in either all types of suicidality or in suicidal ideations. Nevertheless, another meta-analysis found a significant association with suicidal ideation and suicidal attempts, especially with attempts [114]. A meta-analysis that analyzed the relationship between suicide attempts and acute alcohol consumption must be highlighted, finding a significant and very high OR in the case of higher levels of consumption [115]. This may indicate that rather than regular consumption, high and acute alcohol consumption may be the relevant factor for the understanding of suicidality. As for tobacco consumption, other studies found very similar results both in OR value and in the pattern observed according to the suicidality considered [115,116]. Thus, both cited meta-analyses found a higher OR in suicidal attempts than in suicidal ideation, one being performed in people with psychosis [116]. Similar results were found when considering substance use, as another meta-analysis found OR values for suicidal ideation and suicidal attempts close to those obtained by us, as well as a greater OR for suicidal attempts than for suicidal ideation [18]. This same trend was obtained in our meta-analysis and in the meta-analysis of Franklin et al. [44], although a lower OR was found for both suicidality. When considering any mental disorder, all OR were significant and higher than three. This OR value for suicidal attempts was found in a meta-analysis performed on young people [13]. Nevertheless, in the meta-analysis of Franklin et al. [44] the OR obtained for suicidal ideation and suicidal attempts was less than two. Finally, body mass index did not show a significant OR in suicidal ideation, but in suicidal attempts and all types of suicidality it had an OR above four. However, a meta-analysis that uses pooled risk ratios found a significant association with suicidal ideation [24], while no data were found regarding suicidal attempts in the meta-analysis. Two systematic reviews indicate a possible relationship between suicide attempts and a high body mass index [117,118], although one indicates that this relationship only occurred in women [117].

A study of heterogeneity through the analysis moderator shows the influence of the observation period of suicidality in its relation to the factor considered. This analysis was performed mostly for all suicidality due to the greater number of effect sizes. It was observed that in several of the factors at least one of the periods was found to be significant, showing differences compared to the other time periods or at least one period. In demographic factors, a longer period of time had a greater OR in any affective disorder and depression. Nevertheless, of the rest of the clinical factors (anxiety/stress/somatoform disorders, substance use, tobacco use and having any mental disorder), the period of time that obtained the highest OR was the 12-month period. The factors analyzed were gender, educational level, social support, any affective disorder, major depression, anxiety/stress/somatoform disorders, substance use, tobacco use and having any mental disorder. This influence was not observed in relationship status, residential setting or childhood adversity, while the other factors were not analyzed. These differences between the observation period of suicidality have been pointed out by other authors as an aspect to be investigated in order to achieve a greater understanding of suicidality [44,49].

When considering all types of suicidality as a moderator, statistical significance was not reached in any factor. This indicates that the estimated effect of the factors remained stable despite the behavior considered and, therefore, that there are no differences depending on whether specific or other suicidality is considered. This could point in the direction of the consideration of all types of suicidality as a similar concept, although with topographic differences, in which each specific behavior could be seen as a different level of severity [2]. Nevertheless, due to the small number of effect sizes for specific behaviors, we believe that this result should be interpreted with caution. Likewise, it is also possible that this result is due to the lack of standardization in the terminology and instruments used for the collection of information related to suicidality [44,49,50].

Moreover, it seems that several risk factors show distinct associations with suicidality across regions. For instance, there is evidence that the risk factors of residential setting and educational level have different weight when studying suicidal behavior, depending on whether they are studied in the Chinese or the European population. Specifically, the residential setting seems to be relevant only for the Chinese population, whereas the educational level only for the European, as shown by both our meta-analysis and one carried out in China [5]. Additionally, when comparing European with Asian populations, mental illness seems to be a risk factor for the European population but this is not the case for the Asian youth [119]. For this reason, we suggest that future research on suicidality should focus on factors that may be important for specific regions and cultural contexts.

Several limitations must be taken into account in the present study. First, in relation to the search period, in order to update it, for the period between January 1, 2018 to December 31, 2019 we found three articles published in this period on factors associated with suicidality in the general population [120,121,122]. Nevertheless, the only factor in our results that could be modified by this updated search was childhood adversity [121] and a new factor of religiosity that could be included in the analysis [122]. We would like to emphasize, therefore, that the rest of the results would be the same. Secondly, by including only studies published in Spanish or English, this study necessarily presents a publication bias per language, and relevant studies published in languages not considered may be excluded. Thirdly, the informal literature, also called manual or gray, was not included. Therefore, unpublished information is not collected, increased risk of selection and publication bias. Fourth, studies that did not provide enough data for the meta-analysis were excluded, so the results could have varied if they had been included. Fifthly, the limitations of the original studies included in this review should be considered. Although an assessment of the quality of each article has been carried out, obtaining, in general, a moderate quality and studies included were methodologically homogeneous, it should be noted that, in many, cases ad hoc instruments were used for the collection of data related to both factors and suicidality. This may partly explain the high heterogeneity found in the analysis, as well as its high reduction in some cases with the withdrawal of some studies through sensitivity analysis. Sixthly, this heterogeneity can also be explained by the categories created in this study to synthesize and group the original data of the studies (see Appendix A). Moreover, it can be explained by the shortage of effect sizes for certain factors, as shown in Table 2, Table 3 and Table 4. In addition, it should be noted that, since this is not an analysis of experimental studies and mostly cross sectional studies were included, the conclusions about the factors considered here cannot be taken as risk factors [4]. However, this study aims to guide future research in this scientific field by pointing out associations between the factors analyzed. As an eighth limitation, the lack of specifically reviewing and analyzing protective factors against suicide highlighted in different studies should be noted [123,124,125]. Though we acknowledge their importance for suicide research, the inclusion of such protective factors was beyond the scope of this review.

Future research should aim to advance and overcome the difficulties noted by Ribeiro et al. [49]. Our study aims to be a guide in future research by overcoming several of the aspects that these authors point out: the systematization of the type of population included, the homogeneity of methodological designs, geographical areas and cultural factors, the time period of occurrence of suicidality and the description and organization of terminology related to suicidality. Likewise, our study can be used to design clinical guidelines aimed at prevention and action in the case of suicidality.

## 5. Conclusions

This systematic review and meta-analysis conducted in the general European population shows that several factors have a significant association with non-lethal suicidality, with the highest OR for clinical factors, followed by psychosocial factors and, finally, demographic factors. Depression and any affective disorder had the highest OR in all outcomes. Our study tried to overcome several of the aspects that have been identified as difficulties in previous studies: the lack systematization of the type of population included; the heterogeneity of methodological designs, geographical areas and cultural factors; not considering the time period of occurrence of suicidality and the lack of systematization of terminology related to suicidality. Future research is needed in this direction to advance the investigation and prevention of suicidality.

## Figures and Tables

**Figure 1 ijerph-17-04115-f001:**
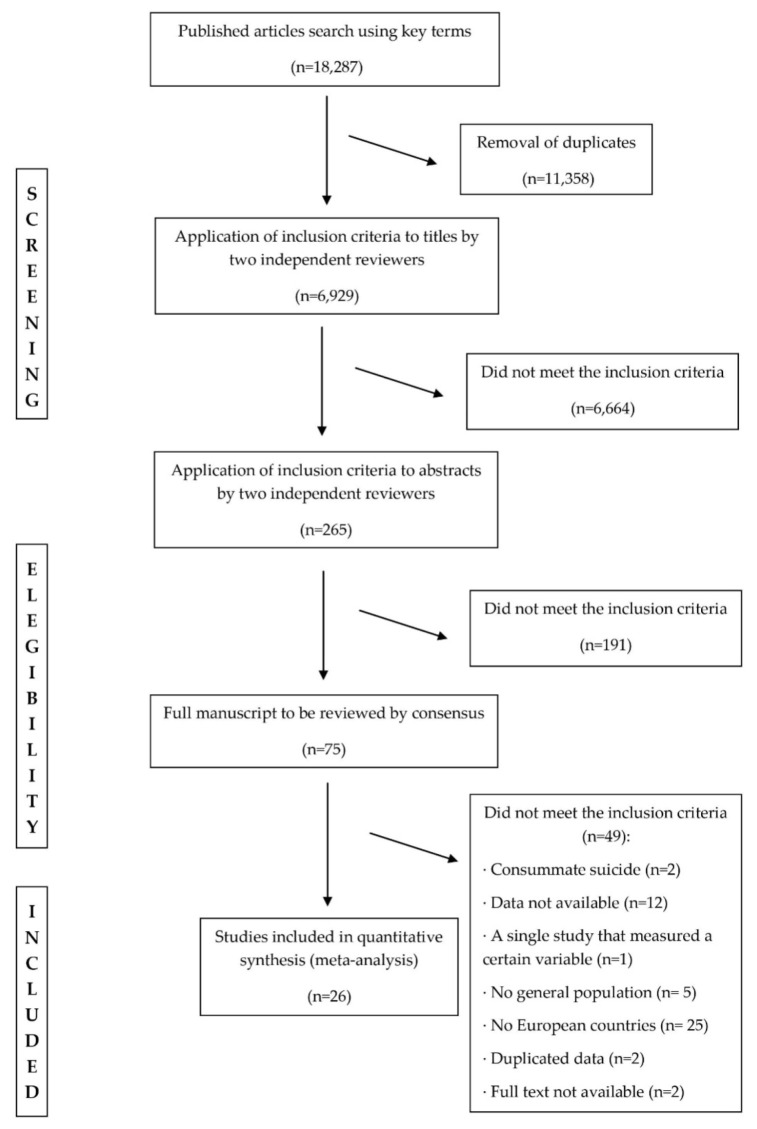
Study selection process.

**Table 1 ijerph-17-04115-t001:** Characteristics and outcome information of included articles.

Authors (Year)	Country ^1^	Age Range	Quality Rating ^2^	Suicidality Type ^3,4^	Period of Time for Suicidality ^4^	Assessment Tools for Demographic Factors ^5^	Assessment Tools for Psychosocial Factors ^5^	Assessment Tools for Clinical Factors ^5^	Factors Analyzed ^4^
Aschan et al. (2013) [60]	England	16–100	Moderate	I, A	Lifetime	Structured interview using computer-assisted personal interviewing	–	CIS-R [61] ^6^ PCL-4 [62] ^7^ SAPAS [63] ^8^ AUDIT [64] ^9^	- Demographic: Age up to 35 years, relationship status, education, employment situation - Clinical: Any mental disorder: anxiety, stress and somatoform disorders (post-traumatic stress disorder), substance use (use of or dependence on drugs), personality dysfunction and grouping of psychological syndromes
Atay et al. (2012) [65]	Turkey	18–65	Moderate	Dw, I, A	Lifetime	Questionnaire ad hoc	Questionnaire ad hoc	SCID-I [66] ^10^	- Demographic: Gender, relationship status, residential setting, education - Psychosocial: Adulthood adversity - Clinical: Family history of mental disorder, any mental disorder: any affective disorder (major depression, dysthymia or persistent depressive disorder, bipolar I and bipolar II), anxiety/stress/somatoform disorders (anxiety disorder not otherwise specified, generalized anxiety disorder, panic disorder, social phobia, specific phobia, adjustment disorder, obsessive-compulsive disorder
Bebbington et al. (2009) [67]	Great Britain	16–74	Moderate	I, A	Point 12-month period Lifetime	Face-to-face interview	Use of cards with options and interview	–	- Demographic: Gender - Psychosocial: Childhood adversity
Blüml et al. (2013) [68]	Germany	18–100	Moderate	A	Lifetime	Face-to-face interview	–	GAD-7 [69] ^11^ PHQ-9 [70] ^12^	- Demographic: Gender, residential setting - Clinical: Any mental disorder: any affective disorder (major depression), anxiety/stress/somatoform disorders (anxiety disorder not otherwise specified)
Boyd et al. (2015) [71]	Belgium Bulgaria France Germany Northern Ireland Italy Portugal Romania Spain The Netherlands	18–100	Moderate	I, P, A	Lifetime	Computer-assisted personal interview Paper-and-pencil interview	–	–	- Demographic: Gender
Bruffaerts et al. (2015) [72]	Belgium	18–100	Moderate	I, A	Lifetime	–	CIDI [73] ^13^	CIDI [73] ^13^	- Psychosocial: Adulthood adversity, childhood adversity, - Clinical: Family history of mental disorder, any mental disorder.
Economou et al. (2016) [74]	Greece	18–79	Moderate	A	Point	Computer-assisted telephone interview	–	SCID-I [66] ^10^	- Demographic: Gender, age up to 35 years, relationship status, residential setting, employment status - Clinical: Any mental disorder: any affective disorder (major depression)
Economou et al. (2013) [75]	Greece	18–79 18–69	Strong	I, A	Point	Computer-assisted telephone interview	–	–	- Demographic: Gender, age up to 35 years, age between 35 and 65 years, age over 65 years, relationship status, employment status
Forkmann et al. (2012) [76]	Germany	14–94	Moderate	I	Point	Demographic data sheet	–	–	- Demographic: Gender, relationship status, residential setting, nationality, employment situation
Gisle & Van Oyen (2013) [77]	Belgium	25–64	Moderate	I, A	Point 12-month period Lifetime	Face-to-face questionnaires	MOS [78] ^14^	–	- Demographic: Gender, employment situation - Psychosocial: Social support
Hintikka et al. (2009) [79]	Finland	25–64	Moderate	I	Point	Questionnaire ad hoc	Questionnaire ad hoc	Questionnaire ad hoc	- Demographic: Gender, age, relationship status, Residential setting, employment situation - Clinical: Family history of mental disorder, any mental disorder: substance use (frequent alcohol consumption and tobacco use)
Hiswåls et al. (2015) [80]	Sweden	16–65	Weak	I	12-month period	Questionnaire ad hoc	Questionnaire ad hoc	Questionnaire ad hoc	- Demographic: Gender, age up to 35 years, age between 35 and 65 years, relationship status, employment, education, employment situation - Psychosocial: Social support - Clinical: Any mental disorder: anxiety/stress/somatoform disorders (anxiety disorder not otherwise specified), substance use (frequent alcohol consumption)
Kovess-Masfety et al. (2011) [81]	France Spain	18–100	Moderate	I, P, A	Lifetime	Computer-assisted personal interview	–	–	- Demographic: Gender
Lara et al. (2015) [82]	Spain	18–100	Moderate	I	12-month period Lifetime	Structured interview using computer-assisted personal interviewing	–	–	- Demographic: Gender
McDonald et al. (2017) [83]	England	16–100	Moderate	I, A	Point 12-month period Lifetime	–	–	Questionnaire ad hoc	- Clinical: Any mental disorder (sleep problems)
Meltzer et al. (2011) [84]	Great Britain	16–100	Moderate	A	Lifetime	–	SLE [85]^15^	–	- Psychosocial: Childhood adversity
Michal et al. (2010) [86]	Germany	35–74	Moderate	I	Point	Computer-assisted personal interview	–	PHQ-9 [70] ^12^ Mini-SPIN [87] ^16^ GAD-7 [69] ^11^ CDS [88] ^17^ DS14 [89] ^18^	- Demographic: Gender, relationship status - Clinical: Any mental disorder: any affective disorder (major depression), anxiety/stress/somatoform disorders (generalized anxiety disorder, panic disorder, social phobia and depersonalization), substance use (frequent alcohol consumption and tobacco use), personality dysfunction. Body mass index
Miret et al. (2014) [90]	Spain	18–100	Strong	I, P, A	12-month period Lifetime	Computer-assisted personal interview	–	–	- Demographic: Gender, age over 65 years, relationship status, residential setting, employment situation
O’Neill et al. (2014) [91]	Northern Ireland	18–100	Moderate	I, P, A	Lifetime	Computer-assisted personal interview	–	–	- Demographic: Gender
Rancāns et al. (2016) [92]	Latvia	18–64	Moderate	Dw, A	12-month period	Computer-assisted personal interview	–	–	- Demographic: Age up to 35 years, gender, relationship status, residential setting, nationality
Saraçli et al. (2016) [93]	Turkey	18–65	Strong	I, A	Lifetime	–	Face-to-face interview Childhood Trauma Questionnaire (CTQ) [94]	BDI [95] ^19^ BAI [96] ^20^	- Psychosocial: Adulthood adversity, childhood adversity - Clinical: Family history of mental disorder, any mental disorder
Scocco et al. (2008) [97]	Italy	15–65	Moderate	I, P, A	Lifetime	Face-to-face interview	–	–	- Demographic: Gender
Spiers et al. (2014) [98]	England Great Britain	16–78	Moderate	I	Point 12-month period	Computer-assisted personal interview	–	–	- Demographic: Age over 65 years
Tempier & Guérin (2015) [99]	England	16–100	Moderate	I, A	12-month period Lifetime	Computer-assisted personal interview	–	Computer-assisted personal interview	- Demographic: Gender - Clinical: Any mental disorder: Substance use (tobacco use)
Ten Have et al. (2013) [100]	The Netherlands	18–64	Moderate	I, P, A	Lifetime	Computer-assisted personal interview	–	–	- Demographic: Gender
Wagner et al. (2013) [101]	Germany	18–100	Moderate	I, A	12-month period Lifetime	–	–	Face-to-face interview	- Clinical: Body mass index

^1^ The following countries were included in meta-analysis: Belgium, Bulgaria, France, Germany, Northern Ireland, Italy, Portugal, Romania, Spain, The Netherlands. ^2^ Study quality according to an adapted version of the Quality Assessment Tool for Quantitative Studies [58]. ^3^ Suicidality type: death wishes (Dw), ideation (I), plan (P), attempts (A). ^4^ Information included in meta-analysis. ^5^ Used to collect information on the factors included in the meta-analysis. ^6^ Clinical Interview Schedule—Revised (CIS-R) [83]. ^7^ Posttraumatic Stress Disorder Checklist (PCL-4) [84]. ^8^ Standardized Assessment of Personality Abbreviated Scale (SAPAS) [85]. ^9^ Alcohol Use Disorders Identification Test (AUDIT) [86]. ^10^ Structured Clinical Interview for DSM-5 (SCID-I) [66]. ^11^ 7-items Generalized Anxiety Disorder (GAD-7) [88]. ^12^ Patient Health Questionnaire (PHQ-9) [89]. ^13^ Composite International Diagnostic Interview (CIDI) [90]. ^14^ Medical Outcome Study (MOS) Social Support Survey [91]. ^15^ Stressful Life Events (SLE) [92]. ^16^ Mini Social Phobia Inventory (Mini-SPIN) [93]. ^17^ Cambridge Depersonalization Scale (CDS) [94]. ^18^ Type-D scale (DS14) [95]. ^19^ Beck Depression Inventory (BDI) [97]. ^20^ Beck Anxiety Inventory (BAI) [98].

**Table 2 ijerph-17-04115-t002:** Calculated odds ratio for all types of suicidality and all time periods.

Factor	*n* ^1^	OR (95% CI) ^2^	*p*-Value	Heterogeneity ^3^
*Demographic factors*				
Gender (woman ^4^)	83	1.56 (1.40–1.72)	<0.05	87.65%
Age up to 35 years	10	1.02 (0.48–2.16)	0.9603	98.52%
Age between 35 and 65 years	5	0.89 (0.39–2.03)	0.7869	98.34%
Age over 65 years	17	0.45 (0.35–0.59)	<0.05	80.85%
Relationship status (Appendix A)	17	0.65 (0.45–0.93)	<0.05	94.90%
Residential setting (urban ^4^)	9	0.69 (0.44–1.10)	0.117	93.89%
Nationality (native ^4^)	3	0.85 (0.73–1.00)	0.0548	0%
Education (university studies ^4^)	4	1.31 (0.22–7.86)	0.7678	98.81%
Employment situation (active ^4^)	16	0.48 (0.32–0.72)	<0.05	95.44%
*Psychosocial factors*				
Social support (low ^4^)	5	2.59 (1.87–3.59)	<0.05	75.53%
Adulthood adversity	6	3.65 (1.94–6.87)	<0.05	86.82%
Childhood adversity	17	3.53 (2.43–5.13)	<0.05	91.12%
*Clinical factors*				
Family history of mental disorder	6	3.03 (1.76–5.23)	<0.05	85.55%
Any affective disorder	9	7.41 (4.13–13.28)	<0.05	84.71%
Major depression	6	7.69 (4.06–14.59)	<0.05	90.63%
Anxiety/stress/somatoform disorders	19	4.29 (2.82–6.51)	<0.05	90.85%
Substance use	15	2.45 (2.01–2.99)	<0.05	84.84%
Frequent alcohol consumption	3	1.52 (0.63–3.70)	0.3527	91.68%
Tobacco use	10	2.67 (2.13–3.34)	<0.05	84.87%
Any mental disorder	57	3.61 (2.90–4.48)	<0.05	94.90%
Body mass index (≥30 Kg/m^2^) ^4^	4	2.58 (1.13–5.89)	<0.05	91.97%

^1^ Number of effect sizes. ^2^ Weighted mean odds ratio with 95% confidence interval. ^3^ Existing amount of heterogeneity between studies measured with I^2^. ^4^ Reference category.

**Table 3 ijerph-17-04115-t003:** Calculated odds ratio for suicidal ideation considering all time periods.

Factor	*n* ^1^	OR (95% CI) ^2^	*p*-Value	Heterogeneity ^3^
*Demographic factors*				
Gender (woman ^4^)	37	1.36 (1.19–1.57)	<0.05	91.19%
Age up to 35 years	4	2.75 (1.15–6.58)	<0.05	98.35%
Age between 35 and 65 years	3	0.51 (0.24–1.07)	0.0759	97.61%
Age over 65 years	10	0.50 (0.39–0.65)	<0.05	74.83%
Relationship status (Appendix A)	8	0.53 (0.36–0.79)	<0.05	93.17%
Residential setting (urban ^4^)	4	0.95 (0.8–1.12)	0.544	0%
Education (university studies ^4^)	3	1.88 (0.19–19.16)	0.5921	99.02%
Employment situation (active ^4^)	9	0.48 (0.31–0.75)	<0.05	94.99%
*Psychosocial factors*				
Social support (low ^4^)	3	2.73 (1.69–4.43)	<0.05	87.77%
Adulthood adversity	3	2.59 (0.95–7.08)	0.064	92.64%
Childhood adversity	7	2.08 (1.13–3.82)	<0.05	91.81%
*Clinical factors*				
Family history of mental disorder	3	2.68 (1.69–4.25)	<0.05	77.80%
Any affective disorder	4	10.95 (4.64–25.82)	<0.05	77.20%
Major depression	3	11.06 (4.09–29.87)	<0.05	89.10%
Anxiety/stress/somatoform disorders	11	5.8 (3.49–9.61)	<0.05	92.71%
Substance use	10	2.18 (1.76–1.19)	<0.05	82.94%
Frequent alcohol consumption	3	1.52 (0.63–3.70)	0.3527	91.68%
Tobacco use	6	2.27 (1.83–2.81)	<0.05	78.71%
Any mental disorder	32	3.90 (2.96–5.13)	<0.05	96.01%
Body mass index (≥30 Kg/m^2^) ^4^	2	2.05 (0.66–6.38)	0.2173	94.05%

^1^ Number of effect sizes. ^2^ Weighted mean odds ratio with 95% confidence interval. ^3^ Existing amount of heterogeneity between studies measured with I^2^. ^4^ Reference category.

**Table 4 ijerph-17-04115-t004:** Calculated odds ratio for suicidal attempts considering all time periods.

Factor	*n* ^1^	OR (95% CI) ^2^	*p*-Value	Heterogeneity ^3^
*Demographic factors*				
Gender (woman ^4^)	28	1.78 (1.46–2.17)	<0.05	79.68%
Age up to 35 years	5	0.67 (0.41–1.09)	0.1027	80.89%
Age between 35 and 65 years	2	2.11 (1.18–3.8)	<0.05	81.47%
Age over 65 years	4	0.42 (0.18–1.01)	0.0516	87.99%
Relationship status (Appendix A)	7	0.71 (0.32–1.56)	0.3939	93.42%
Residential setting (urban ^4^)	4	0.60 (0.22–1.66)	0.3249	96.07%
Employment situation (active ^4^)	7	0.48 (0.21–1.09)	<0.05	95.08%
*Psychosocial factors*				
Social support (low ^4^)	2	2.27 (1.63–3.17)	<0.05	0%
Adulthood adversity	3	5.52 (2.89–10.52)	<0.05	58.93%
Childhood adversity	10	5.45 (4.04–7.35)	<0.05	67.60%
*Clinical factors*				
Any affective disorder	4	6.04 (1.84–21.06)	<0.05	86.24%
Major depression	3	7.09 (2.19–22.93)	<0.05	89.70%
Anxiety/stress/somatoform disorders	4	3.15 (1.50–6.65)	<0.05	63.37%
Substance use	5	3.26 (2.32–4.60)	<0.05	73.58%
Tobacco use	4	3.62 (2.46–5.34)	<0.05	73.28%
Any mental disorder	19	3.24 (2.14–4.92)	<0.05	93.15%
Body mass index (≥30 Kg/m^2^) ^4^	1	4.23 (2.57–6.96)	<0.05	0%

^1^ Number of effect sizes. ^2^ Weighted mean odds ratio with 95% confidence interval. ^3^ Existing amount of heterogeneity between studies measured with I^2^. ^4^ Reference category.

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
