# Peer review of "Determinants of Suicidality in the European General Population: A Systematic Review and Meta-Analysis"

_ijerph, 2020, doi:10.3390/ijerph17114115_

Round 1
Reviewer 1 Report
This was challenging to read, perhaps because of the topic but the methods process was complex. Reducing down to such a small number of final studies seemed like too much and having assessed purpose and tables in particular I understand the thought process and recommend for final steps in acceptance process.
Author Response
Thank you very much for your interest and recommendations.
Reviewer 2 Report
I have the following comments for the authors to address.
1) The authors should mention a recent study published by IJERPH. The studies focused on children and adolescents. Please this as special population:
These studies include systematic reviews and/or meta-analyses that have
focused on investigating suicidality in the general population [5]; in certain specific populations as, for example, children and adolescents (Lim et al 2019), students [6], prisoners [7] and inpatients [8].
Reference:
Lim KS, Wong CH, McIntyre RS, et al. Global Lifetime and 12-Month Prevalence of Suicidal Behavior, Deliberate Self-Harm and Non-Suicidal Self-Injury in Children and Adolescents between 1989 and 2018: A Meta-Analysis. Int J Environ Res Public Health. 2019;16(22):4581. Published 2019 Nov 19. doi:10.3390/ijerph16224581
2) Why did the authors include the years of 2008 and 2017 only? What is the scientific reason or a good reason? Ideally, they should include studies from 2000 after EU adopted the same currency and EU became more untied than in 1990s and end in 2019 before Brexit. The authors need to explain under methodology.
3) Why the authors did not calculate the pooled lifetime prevalence of suicidal plan and suicide attempts? Please refer to the method of the following study. They should calculate the pooled lifetime prevalence of suicidal plan and attempts and compare with the following study.
Lim KS, Wong CH, McIntyre RS, et al. Global Lifetime and 12-Month Prevalence of Suicidal Behavior, Deliberate Self-Harm and Non-Suicidal Self-Injury in Children and Adolescents between 1989 and 2018: A Meta-Analysis. Int J Environ Res Public Health. 2019;16(22):4581. Published 2019 Nov 19. doi:10.3390/ijerph16224581
4) Under discussion, the authors should discuss how factors associated with suicide in Europeans similar or different from Asians. I recommend the authors to search for Asian landmark papers by entering the following search terms into Pubmed: Insight Into Prescipitants for Asian Suicide Attempters [ti]
5) Under limitations, the authors should add the seventh limitation. This study did not calculate OR for protective factors. Please mention common protective factors that should be included and require future study. Please search for a landmark study on protective factors against suicide by entering the following search terms into Pubmed: Suicide Attempts in a Multi-Ethnic Asian Society [ti]
Round 2
Reviewer 2 Report
I recommend publication.